# The diagnostic tests and functional outcomes of acute ischemic stroke or transient ischemic attack in young adults: A 4-year hospital-based observational study

Jakraphat Sakseranee[1☉], Peerapong Sethabouppha[1☉], Tanapat Pattarasakulchai[1☉], Theerawee Klaewkla[1☉], Kitti Thiankhaw[2,3]*

1 Faculty of Medicine, Chiang Mai University, Chiang Mai, Thailand, 2 Division of Neurology, Department of Internal Medicine, Faculty of Medicine, Chiang Mai University, Chiang Mai, Thailand, 3 The Northern Neuroscience Center, Faculty of Medicine, Chiang Mai University, Chiang Mai, Thailand

☉ These authors contributed equally to this work.
* kitti.th@cmu.ac.th

## Abstract

### Background and objectives

Ischemic strokes in young adults have been a significant concern due to various potential etiologies and had substantial clinical and public health impacts. We aimed to study the diagnostic tests, etiologies, and functional outcomes of acute ischemic stroke (AIS) and transient ischemic attack (TIA) in young adult patients.

### Methods

The data were retrieved from the Chiang Mai University Hospital Stroke Registry between January 2018 and December 2021. Consecutive AIS or TIA patients were included if they were 18–50 years and had no stroke mimics. Study outcomes were proportions of positive diagnostic tests, and 90-day modified Rankin Scale (mRS).

### Results

Of 244 enrolled patients, 59.0% (n = 144) were male, and 38.1% (n = 93) were aged 18–40, classified as the younger age group. There was a high incidence of diabetes (24.5%) and dyslipidemia (54.3%) among patients aged 41–50, associated with small-vessel occlusion and large-artery atherosclerosis stroke classification in this age group. Patients aged 18–40 years had more other determined etiologies (39.8%), with hypercoagulability (8.2%), arterial dissection (7.8%), and cardiac sources (6.6%) being the first three causes, which were associated with higher anticoagulant treatment. The cerebrovascular study, cardiac evaluation using echocardiography, and antiphospholipid syndrome testing were commonly performed, of which computed tomography angiography provided a high proportion of positive results (80.3%). 76.3% of young adult patients had excellent functional outcomes (mRS 0–1) with a median mRS of 0 (interquartile range 0–1) at 90-day follow-up.

**Data Availability Statement:** Data cannot be shared publicly because of ethical issues. Data are

available from the Ethics Committee of the Faculty of Medicine, Chiang Mai University (contact via researchmed@cmu.ac.th) for researchers who meet the criteria for access to confidential data.

**Funding:** The author(s) received no specific funding for this work.

**Competing interests:** The authors have declared that no competing interests exist.

## Conclusions

Stroke of other determined etiology remained the common cause of stroke in young adults, and most affected individuals had excellent clinical outcomes. Blood tests for arterial hypercoagulability and noninvasive vascular and cardiac evaluations are encouraged in selected patients to determine the stroke etiology and guide for appropriate preventive strategies.

## Introduction

Strokes in young adult patients are considered not uncommon, accounting for about 15% of all stroke patients [1]. It significantly affects the quality of life and has high economic impacts since affected individuals might become disabled before their most productive years, in contrast to strokes in older persons [2]. This background makes preventing stroke in young adults an urgent global public health concern. Because the etiologic spectrum of ischemic stroke in young adults is broader and more heterogeneous than in older people, [3] several investigations and tests are usually performed to evaluate all potential etiologies to select appropriate secondary stroke prevention, including standard, advanced, and specialized evaluation [4].

Previous studies have been conducted to elucidate the risk factors, stroke outcomes and its predictors, and causes of stroke in young adult patients with the general agreement that uncommon causes and embolic sources of stroke might be the potential causes in individuals with younger age [5–7]. Importantly, there is no consensus on the definition of stroke in the young, and these studies ranged in age from 40 to 60 while determining young adults. Additionally, studies focused on different causes between gender and compared among young and younger adults are limited.

In the present study, we aimed to study the diagnostic tests of acute ischemic stroke (AIS) and transient ischemic attack (TIA) in young adults. We also investigated the causes and functional outcomes of stroke in this population using a large cohort of consecutive young adults with AIS or TIA. We hypothesized that our findings probably guided effective diagnostic methods to reduce the number of unnecessary investigations and establish appropriate stroke prevention.

## Materials and methods

### Study design and population

We conducted a single-center retrospective study from Maharaj Nakorn Chiang Mai Hospital. The study received approval from the Institutional Review Board, Faculty of Medicine, Chiang Mai University, Chiang Mai, Thailand, study code: MED-2565-08769. All data from this study were fully anonymized before access, and the Research Ethics Committee waived the requirement for informed consent. Data were retrieved from the Chiang Mai University Hospital Stroke Registry, which prospectively collected consecutive patients diagnosed with all types of acute stroke. Acute ischemic stroke and TIA patients aged 18 to 50 between January 2018 and December 2021 were enrolled in the study. AIS was defined as a clinical presentation of sudden or acute onset of focal neurological deficits with brain imaging-confirmed lesions, whereas TIA had no symptomatic lesions on the brain imaging. Patients with acute hemorrhagic stroke or finally diagnosed with stroke mimics were excluded. We further divided the eligible study population into two cohorts, gender, male or female, and age, cut point of 40 years (Fig 1).

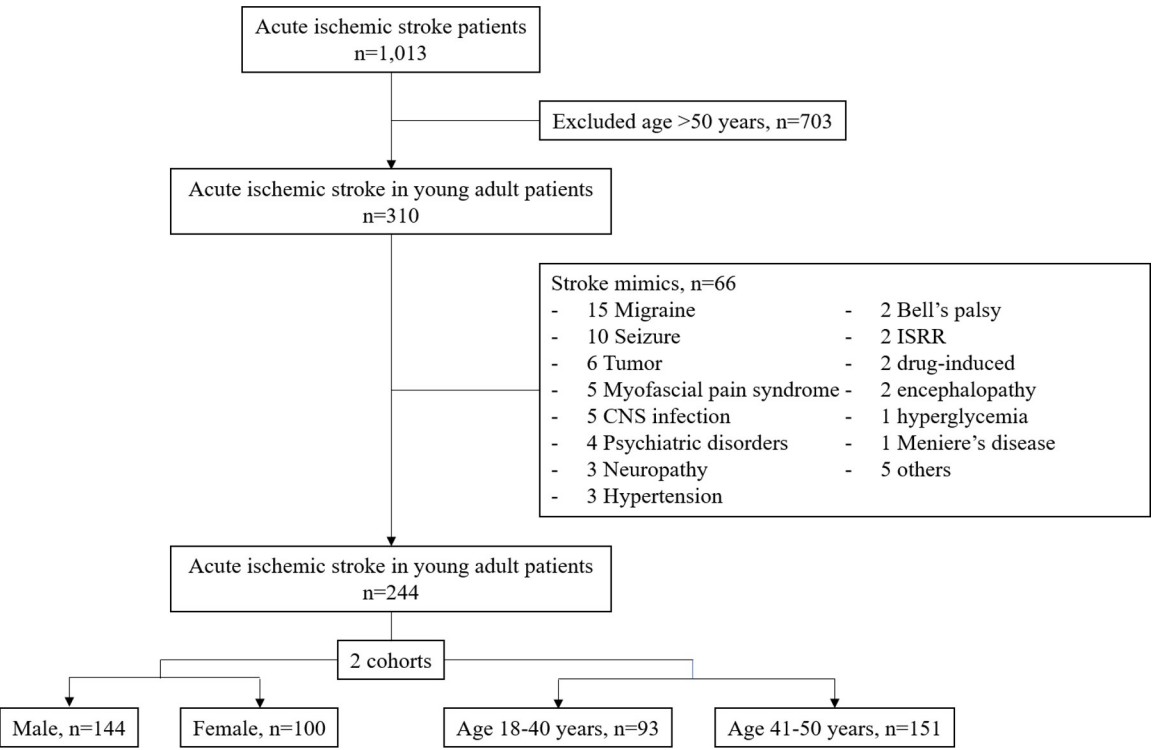

**Fig 1. Study flow chart and patient selection procedure.** CNS, central nervous system; ISRR, Immunization Stress Related Response.

## Data collection and outcomes

The demographic data, medical history, risk factors, and laboratory results were systematically collected from the electronic medical record (EMR). A primary outcome of the study was the diagnostic tests performed on young adult patients. These tests included standard diagnostic tests for AIS patients, anti-human immunodeficiency virus (HIV), erythrocyte sedimentation rate (ESR), antinuclear antibody (ANA) titer, arterial and venous hypercoagulability tests (lupus anticoagulant (LA), anticardiolipin antibody (aCL), anti-β2-glycoprotein antibody (anti-β2GP), protein C and S, and antithrombin III), and genetic testing. The final results of the standard and advanced blood tests were classified as normal (negative) and abnormal (positive) tests based on the standardized laboratory reference ranges. The cardiac evaluation and cerebrovascular study results were based on the official report from a certified board cardiologist and a certified board neuroradiologist, respectively, and were blinded to the study outcomes.

Stroke parameters and outcomes were reviewed and justified by a stroke neurologist (KT). The Trial of Org 10172 in Acute Stroke Treatment (TOAST) classification of subtypes of acute ischemic stroke was applied for ischemic stroke classification [8] and neurological outcomes, including the National Institute of Health Stroke Scale (NIHSS), modified Rankin Scale (mRS), and Barthel Index (BI) was utilized to define the functional outcomes at admission and 90 days. 90-day mRS 0–1 was categorized as an excellent outcome, and 0–2 as a favorable functional outcome. Causes of ischemic stroke of other determined etiology were based on the final diagnosis given by the attending neurologists or the International Classification of Diseases 10th Revision (ICD-10) code recorded, and some conditions used scientific evidence such as a high Risk of Paradoxical Emboli (RoPE) score for patent foramen ovale (PFO)-

related stroke and genetic testings for mitochondrial disease (Mitochondrial Encephalopathy, Lactic Acidosis, and Stroke-like episodes or MELAS) and monogenic stroke disorders (NOTCH3 mutation for Cerebral Autosomal Dominant Arteriopathy with Subcortical Infarcts and Leukoencephalopathy or CADASIL).

## Statistical analysis

Categorical variables were presented in number and proportion (%), and comparisons between groups were performed using the Pearson $\chi^2$ test or Fisher exact 2-sided test as appropriate. Continuous variables were displayed as mean with standard deviation (SD) or median with the interquartile range (IQR) depending on data distribution. Comparisons between groups of continuous variables were performed using a two-tailed Student $t$-test for parametric data and Mann-Whitney U (Wilcoxon rank sum) test for non-parametric data. The odds ratio (OR) and corresponding 95% confidence interval (CI) were used to evaluate conventional risk factors and ischemic stroke subtypes, and statistical significance was indicated when a two-sided $P$ value was less than 0.05. All statistical analyses were performed using licensed Stata statistical software version 16.1 (Stata Statistical Software: Release 16.1, Stata Corporation, College Station, TX, 2019).

## Results

### Stroke mimics and effects of gender and age group

Three hundred and ten young adult patients with acute ischemic stroke aged 18–50 years were initially enrolled. Of these, 66 patients (21.3%) were diagnosed with stroke mimics and excluded from the study. Migraine was the most common stroke mimic in our cohort, 15 patients, followed by seizure and tumor, ten and six patients, respectively (Fig 1). Of the remaining 244 patients, 144 (59.0%) were male, and 93 (38.1%) were younger age group, 18–40 years. Among gender cohorts, male patients had higher body mass index (BMI) and current smoking (41% vs. 5.0%, $P < 0.001$), and alcohol drinking (48.6% vs. 12.0%, $P < 0.001$). AIS patients in a younger group had a higher proportion of oral contraceptive pills (OCP) used (6.5% vs. 0%, $P = 0.003$), while patients aged 41–50 group had significantly higher vascular risk factors or comorbidities, including diabetes mellitus, and dyslipidemia (24.5% vs. 10.8%, $P = 0.008$ and 54.3% vs. 38.7%, $P = 0.02$, respectively). Interestingly, congestive heart failure (CHF) was commonly diagnosed in the younger patient group, in which non-ischemic dilated cardiomyopathy was a certain cause in this population (10.8% vs. 1.3%, $P = 0.001$). Table 1 demonstrates the baseline characteristics and risk factors of the participants, classified into two cohorts.

### Stroke parameters and outcomes, risk factors, and causes of other determined stroke etiology

Table 2 illustrates the routine laboratory results and stroke variables among the two age groups of patients. Compared to a younger age group, AIS aged 41–50 had more prevalence of large-artery atherosclerosis (LAA) and small-vessel occlusion (SVO) stroke subtype according to TOAST classification, 21.9% vs. 9.7%, $P = 0.014$ and 27.8% vs. 11.8%, $P = 0.003$, respectively (Fig 2). These corresponded with the ORs of risk factors by ischemic stroke subtype (Table 3), ORs of hypertension and LAA is 3.09 (95% CI 1.36–7.01, $P < 0.05$). Diabetes mellitus is associated with LAA and SVO subtypes of AIS, and dyslipidemia for cardioembolism (CE), SVO, and stroke of other determined etiology see Table 3.

**Table 1. Baseline characteristics and risk factors of the patients.**

| Characteristics | Total cohort, No. (%) (n = 244) | Gender, No. (%) | | P value | Age, No. (%) | | P value |
|---|---|---|---|---|---|---|---|
| | | Male (n = 144) | Female (n = 100) | | 18–40 yr (n = 93) | 41–50 yr (n = 151) | |
| Demographics | | | | | | | |
| Age, yr–mean (SD) | 41.0 (8.2) | 41.2 (7.9) | 40.9 (8.7) | 0.80 | 32.2 (5.8) | 46.5 (3.0) | <0.001 |
| Sex | NA | 144 (59.0) | 100 (41.0) | NA | 54 (58.1) | 90 (59.6) | 0.81 |
| BMI, kg/m$^2$ –mean (SD) | 25.1 (5.1) | 25.9 (5.2) | 24.1 (4.9) | 0.02 | 25.7 (4.9) | 24.7 (5.3) | 0.21 |
| SBP, mmHg–mean (SD) | 148.8 (32.7) | 151.2 (33.4) | 145.5 (31.5) | 0.18 | 144.7 (31.9) | 151.4 (33.0) | 0.12 |
| Heart rate, bpm–mean (SD) | 88.3 (16.4) | 87.0 (15.7) | 90.2 (17.2) | 0.13 | 89.4 (15.4) | 87.6 (17.0) | 0.39 |
| Admission NIHSS–median (IQR) | 3 (1–6.5) | 3 (2–8) | 2 (1–5) | 0.09 | 3 (2–7) | 3 (1–6) | 0.44 |
| Admission mRS–median (IQR) | 2 (1–4) | 2 (1.5–4) | 2 (1–4) | 0.18 | 2 (1–3) | 2 (1–4) | 0.38 |
| Medical history and risk factors | | | | | | | |
| Hypertension | 105 (43.0) | 66 (45.8) | 39 (39.0) | 0.29 | 33 (35.5) | 72 (47.7) | 0.06 |
| Diabetes mellitus | 47 (19.3) | 30 (20.8) | 17 (17.0) | 0.46 | 10 (10.8) | 37 (24.5) | 0.008 |
| Dyslipidemia | 118 (48.4) | 78 (54.2) | 40 (40.0) | 0.03 | 36 (38.7) | 82 (54.3) | 0.02 |
| Prior stroke/TIA | 40 (16.4) | 22 (15.3) | 18 (18.0) | 0.57 | 13 (14.0) | 27 (17.9) | 0.42 |
| VHD | 10 (4.1) | 7 (4.9) | 3 (3.0) | 0.53 | 3 (3.2) | 7 (4.6) | 0.75 |
| CAD | 5 (2.1) | 3 (2.1) | 2 (2.0) | 1.00 | 2 (2.2) | 3 (2.0) | 1.00 |
| Atrial fibrillation | 14 (5.7) | 6 (4.2) | 8 (8.0) | 0.21 | 3 (3.2) | 11 (7.3) | 0.26 |
| CHF | 12 (4.9) | 9 (6.3) | 3 (3.0) | 0.37 | 10 (10.8) | 2 (1.3) | 0.001 |
| Cancer | 9 (3.7) | 3 (2.1) | 6 (6.0) | 0.17 | 1 (1.1) | 8 (5.3) | 0.16 |
| Headache | 11 (4.5) | 5 (3.5) | 6 (6.0) | 0.37 | 3 (3.2) | 8 (5.3) | 0.54 |
| Pregnancy/puerperium | 2 (0.8) | NA | 2 (2.0) | NA | 2 (2.2) | 0 | 0.38 |
| OCP used | 6 (2.5) | NA | 6 (6.0) | NA | 6 (6.5) | 0 | 0.003 |
| Illicit drug used | 9 (3.7) | 8 (5.6) | 1 (1.0) | 0.09 | 6 (6.5) | 3 (2.0) | 0.09 |
| Vaccination | 5 (2.1) | 4 (2.8) | 1 (1.0) | 0.65 | 1 (1.1) | 4 (2.7) | 0.65 |
| Current smoking | 64 (26.2) | 59 (41.0) | 5 (5.0) | <0.001 | 22 (23.7) | 42 (27.8) | 0.76 |
| Current alcohol drinking | 82 (33.6) | 70 (48.6) | 12 (12.0) | <0.001 | 33 (35.5) | 49 (32.5) | 0.75 |

**Abbreviations:** BMI, body mass index; CAD, coronary artery disease; CHF, congestive heart failure; IQR, interquartile range; mRS, modified Rankin Scale; NA, not applicable; NIHSS, National Institute of Health Stroke Scale; OCP, oral contraceptive pills; SBP, systolic blood pressure; SD, standard deviation; TIA, transient ischemic attack; VHD, valvular heart disease.

Focus on younger patients, AIS patients in the 18–40 age group had a significantly higher proportion of stroke of other determined etiology, 39.8% vs. 25.2%, P = 0.016, compared to an older group (Table 2 and Fig 2). The most common cause of ischemic stroke of other determined etiology is a hypercoagulable state (8.2%), and hereditary thrombophilia is commonly diagnosed (4.1%). These findings are associated with a higher proportion of anticoagulant prescriptions in AIS patients in the younger age group (23.7% vs. 13.4%, P = 0.04). The following etiologies included vascular dissection (7.8%), in which anterior circulation at the internal carotid artery (ICA) is more prevalent than those in posterior circulation dissection. Cardiac sources of stroke go to the third rank at 6.6%, and PFO with or without atrial septal aneurysm (ASA) based on the RoPE score and echocardiographic findings accounts for a common cause among cardiac etiology. Table 4 shows the cause of ischemic stroke of other determined etiology among the participants.

AIS patients in younger age groups tend to have more excellent or favorable functional outcomes at 90 days, based on the NIHSS score, mRS, and BI, although there was no statistical significance (Fig 3). There was no significant difference between the two groups in routine laboratory results, reperfusion therapy, and complications following treatment (Table 2).

**Table 2. Laboratory results and stroke parameters of the patients.**

| Parameters | Total cohort (n = 244) | 18–40 yr (n = 93) | 41–50 yr (n = 151) | *P* value |
|---|---|---|---|---|
| Laboratory results | | | | |
| Hemoglobin, g/dL–mean (SD) | 13.3 (2.6) | 13.3 (2.3) | 13.3 (2.7) | 0.87 |
| Platelet count, cells/μL–mean (SD) | 288,359 (123,610) | 295,652 (160,581) | 283,886 (94,405) | 0.47 |
| Fasting blood glucose, mg/dL–median (IQR) | 97 (87–122) | 95 (85–117.5) | 97 (90–126) | 0.06 |
| Hemoglobin A1C, %–median (IQR) | 5.78 (5.38–7.61) | 5.60 (5.26–6.54) | 5.90 (5.45–8.20) | 0.07 |
| HDL-C, mg/dL–mean (SD) | 45.2 (13.5) | 46.2 (13.8) | 44.6 (13.4) | 0.42 |
| LDL-C, mg/dL–mean (SD) | 130.8 (55.1) | 139.0 (65.2) | 126.4 (48.4) | 0.11 |
| Ischemic stroke classification–no. (%) | | | | |
| Large-artery atherosclerosis | 42 (17.2) | 9 (9.7) | 33 (21.9) | 0.014 |
| Cardioembolism | 24 (9.8) | 13 (14.0) | 11 (7.3) | 0.09 |
| Small-vessel occlusion | 53 (21.7) | 11 (11.8) | 42 (27.8) | 0.003 |
| Stroke of other determined etiology | 75 (30.7) | 37 (39.8) | 38 (25.2) | 0.016 |
| Stroke of undetermined etiology | 50 (20.5) | 23 (24.7) | 27 (17.9) | 0.20 |
| Treatment–no, (%) | | | | |
| Intravenous thrombolysis | 29 (11.9) | 14 (15.1) | 15 (9.9) | 0.23 |
| Mechanical thrombectomy | 1 (0.4) | 0 | 1 (0.7) | 1.00 |
| Antiplatelet | 187 (77.3) | 66 (71.0) | 121 (81.2) | 0.10 |
| Anticoagulant | 42 (17.4) | 22 (23.7) | 20 (13.4) | 0.04 |
| Steroid/immunosuppressive drugs | 5 (2.1) | 3 (3.2) | 2 (1.3) | 0.16 |
| Antibiotics | 5 (2.1) | 0 | 5 (3.4) | 0.37 |
| Complications | | | | |
| Asymptomatic ICH | 6 (2.5) | 1 (1.1) | 5 (3.3) | 0.41 |
| Symptomatic ICH | 2 (0.8) | 1 (1.1) | 1 (0.7) | 1.00 |
| Cerebral edema | 16 (6.6) | 7 (7.5) | 9 (6.0) | 0.63 |
| 90-day functional outcomes | | | | |
| NIHSS–median (IQR) | 0 (0–1) | 0 (0–1) | 0 (0–1) | 0.96 |
| mRS–median (IQR) | 0 (0–1) | 0 (0–1) | 0 (0–2) | 0.58 |
| mRS 0–1 –no. (%) | 174 (76.3) | 71 (81.6) | 103 (73.1) | 0.14 |
| mRS 0–2 –no. (%) | 197 (86.4) | 79 (90.8) | 118 (83.7) | 0.13 |
| BI–median (IQR) | 100 (90–100) | 100 (90–100) | 100 (90–100) | 0.95 |

**Abbreviations:** BI, Barthel Index; HDL-C, high-density lipoprotein cholesterol; ICH, intracerebral hemorrhage; IQR, interquartile range; LDL-C, low-density lipoprotein cholesterol; mRS, modified Rankin Scale; NIHSS, National Institute of Health Stroke Scale; SD, standard deviation.

## Diagnostic test results among the participants

Apart from the standard or routine blood tests, noninvasive vascular study with computed tomography angiography (CTA) is commonly performed in young adults with AIS, representing 80.3% of positive tests. Male patients had 12.7% of hemoglobin more than 16.5 g/dL, compared to 1.0% in females, $P < 0.001$, while female patients had more positive tests for an autoimmune-related condition, including ESR more than 20 mm/hr and positive ANA titer, 48.2% vs. 19.2%, $P = 0.04$, and 35.3% vs. 8.3%, $P = 0.03$, respectively. Among cardiac evaluation, transthoracic echocardiography (TTE) was commonly performed in 44.7% (109 of 244) of patients and provided positive results in 32.1% with no difference between gender or age groups. Transesophageal echocardiography (TEE) illustrated diagnostic results in 60% of included patients. Still, its diagnostic yield cannot be generalized clinical implication because it was performed on only ten patients and selected populations with highly suspected positive cardiac causes. Among advanced blood tests for hereditary thrombophilia, tests for arterial

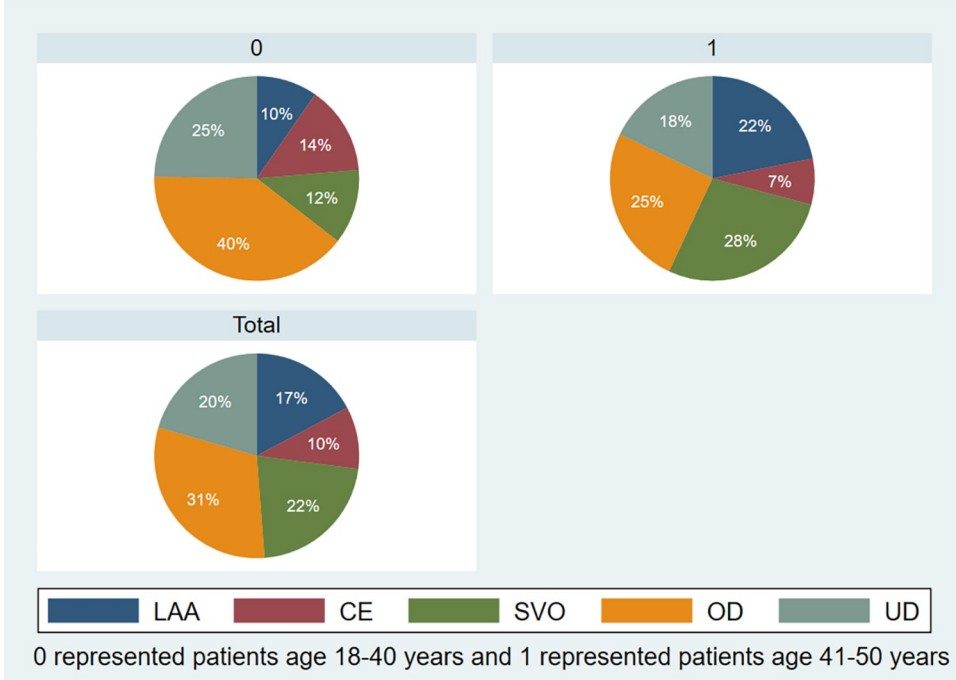

**Fig 2. TOAST classification of subtype of acute ischemic stroke.** A) age 18–40 years, B) age 41–50 years, C) total cohort. CE, cardioembolism; LAA, large-artery atherosclerosis; OD, stroke of other determined etiology; SVO, small-vessel occlusion; UD, stroke of undetermined etiology.

thrombosis or antiphospholipid syndrome (APS), including LA and aCL is usually performed, but anti-β2GP is a higher proportion of positive result at 18.5%. Table 5 and S1 Fig demonstrate the diagnostic test results of ischemic stroke in young adults.

## Discussion

As the leading cause of long-term disability, especially in young adults, stroke significantly contributes to the high cost of health care. Because of incremental disabilities with the episodes of stroke recurrence, the current approach to patients with newly diagnosed strokes is to

**Table 3. Odds ratio of risk factors by ischemic stroke subtype.**

| Risk factors | Ischemic stroke subtype, OR (95% CI) | | | | |
|---|---|---|---|---|---|
| | **LAA** | **CE** | **SVO** | **OD** | **UD** |
| Hypertension | 3.09* (1.36–7.01) | 1.70 (0.65–4.44) | 1.86 (0.91–3.79) | 0.43* (0.22–0.85) | 0.37* (0.17–0.84) |
| Diabetes mellitus | 5.32** (2.43–11.62) | 0.72 (0.19–2.73) | 2.41* (1.15–5.04) | 0.17* (0.05–0.59) | NA |
| Dyslipidemia | 1.12 (0.49–2.54) | 0.29* (0.10–0.82) | 3.58** (1.67–7.67) | 0.50* (0.26–0.96) | 1.33 (0.64–2.77) |
| Prior stroke/TIA | 1.06 (0.40–2.78) | 2.05 (0.72–5.86) | 0.54 (0.21–1.40) | 2.55* (1.15–5.66) | 0.16* (0.04–0.72) |
| Current smoking | 1.07 (0.64–1.80) | 0.77 (0.42–1.42) | 1.11 (0.70–1.75) | 0.83 (0.52–1.31) | 1.25 (0.74–2.10) |
| Current alcohol drinking | 1.08 (0.66–1.75) | 1.70 (0.97–3.00) | 1.04 (0.68–1.61) | 0.88 (0.57–1.34) | 0.75 (0.46–1.24) |

**Abbreviations:** CAD, coronary artery disease; CE, cardioembolism; CI, confidence interval; LAA, large-artery atherosclerosis; NA, not applicable; OD, a stroke of other determined etiology; OR, Odds ratio; SVO, small-vessel occlusion; TIA, transient ischemic attack; UD, a stroke of undetermined etiology; VHD, valvular heart disease.

* $P$ value < 0.05

** $P$ value < 0.001

**Table 4. Causes of ischemic stroke of other determined etiology.**

| Causes (n = 75) | No. (%) |
|---|---|
| Cardiac sources of stroke | 16 (6.6) |
| PFO and/or ASA | 5 (2.0) |
| Cardiomyopathy | 3 (1.2) |
| LA/LV thrombus | 3 (1.2) |
| IE | 5 (2.0) |
| Vascular dissection | 19 (7.8) |
| ICA dissection | 10 (4.1) |
| VA/BA dissection | 9 (3.7) |
| Hypercoagulable state | 20 (8.2) |
| APS | 5 (2.0) |
| Hereditary thrombophilia | 10 (4.1) |
| Cancer-related | 5 (2.0) |
| Vasculopathy | 12 (4.9) |
| Vasculitis | 5 (2.0) |
| RCVS | 1 (0.4) |
| Moyamoya disease | 2 (0.8) |
| HIV | 4 (1.6) |
| Genetic diseases | 3 (1.2) |
| CADASIL | 2 (0.8) |
| MELAS | 1 (0.4) |
| MPN | 2 (0.8) |
| Drug-induced | 1 (0.4) |
| Other causes* | 2 (0.8) |

**Abbreviations:** APS, antiphospholipid syndrome; ASA, atrial septal aneurysm; BA, basilar artery; CADASIL, Cerebral Autosomal Dominant Arteriopathy with Sub-cortical Infarcts and Leukoencephalopathy; HIV, human immunodeficiency virus; ICA, internal carotid artery; IE, infective endocarditis; LA, left atrium; LV, left ventricle; MELAS, Mitochondrial Encephalopathy, Lactic Acidosis, and Stroke-like episodes; MPN, myeloproliferative neoplasms; PFO, patent foramen ovale; RCVS, reversible cerebral vasoconstriction syndrome; VA, vertebral artery.
* Other causes included one hemodynamic stroke and one pregnancy-related stroke.

establish the etiology or mechanism of the stroke to optimize the prevention of secondary ischemic stroke [9, 10]. Based on the current recommendations and guidelines for the prevention of stroke, extensive algorithmic approaches with advanced and specialized evaluation are performed in young adult individuals, however, the diagnostic yield of these tests is still debated [4]. The study aimed to investigate the diagnostic tests carried out in young adult AIS patients, functional outcomes, and causes of stroke found in this age group. Besides standard evaluation or basic laboratory tests, anti-HIV, TTE, hypercoagulable state, especially APS screening, and noninvasive diagnostic vascular study with CTA are usually performed. On the contrary, the diagnostic yields of these investigations, especially compared to others, should be interpreted cautiously because it was not entirely performed in all cohorts.

Diagnostic testing for stroke origin centers on reducing the potential for stroke recurrence. The tests listed in this study focus on all domains. The specialized investigation, genetic testing for CADASIL (NOTCH3 mutation) and MELAS is often conducted in young adult patients with clinical clues compatible with these stroke syndromes. Additionally, these diagnostic procedures are usually done as a step-by-step approach in patients who have undergone all other investigations yet have not found any etiology, which only included three patients in our

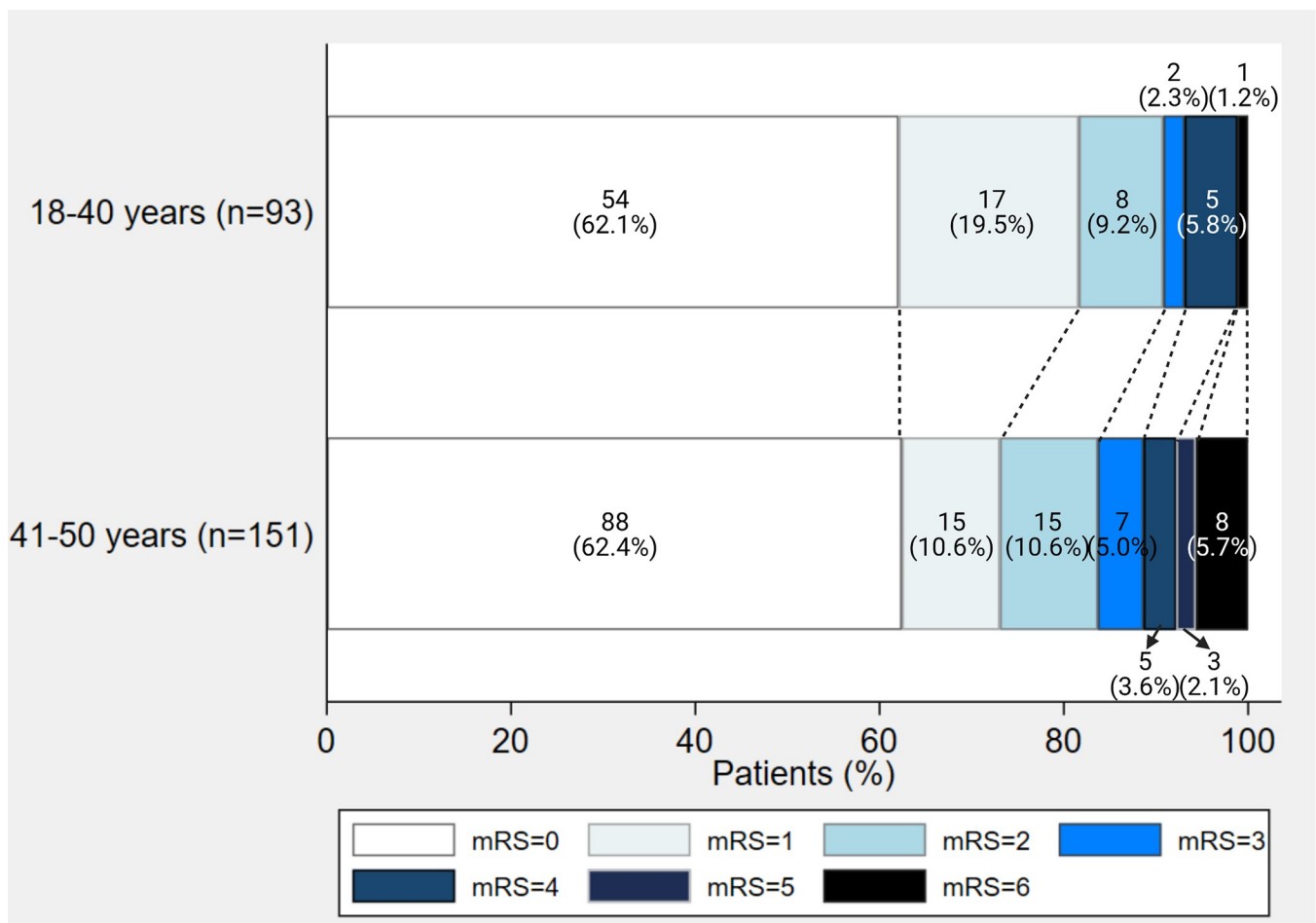

**Fig 3. Functional outcomes of ischemic stroke in young adults.** mRS, modified Rankin Scale.

study. As a result, the value of these diagnostic yields typically varied between the studies. Dong Y.'s study conducted in two hundred eighteen consecutive patients found the overall frequency of CADASIL mutations in lacunar stroke to be less than 0.5% [11]. Therefore, the difference in the suspected population might affect the inconsistent results of the diagnostic yield of genetic tests in young individuals who suffer from acute ischemic stroke or TIA.

The vascular study of cerebral vessels and neck, including CTA, magnetic resonance angiography (MRA), or carotid doppler ultrasonography (CDUS), provides details of cerebral vasculatures and guides therapeutic interventions in acute stroke settings and long-term secondary stroke prevention [12, 13]. Therefore, noncontrast computed tomography (NCCT) and CTA brain have been utilized as standard stroke neuroimaging in acute stroke patients [14]. Among these investigations, our finding reveals that CTA represents a higher proportion of positive results compared to MRA and CDUS. Limitations and accessibility of MRA and operator-dependent of CDUS might be the answer for our cohort that these vascular imagings are lower number performed and proportion of positive tests. Similar to Ji R. *et al.*, which also found a superiority of vascular imaging, CTA, in young individuals [15]. According to the 2021 Guidelines for the Prevention of Stroke in Patients with Stroke and Transient Ischemic Attack: A Guideline From the American Heart Association/American Stroke Association, the lesion is first determined to be an ischemic stroke using CT. After that, CTA is then carried

**Table 5. Diagnostic test results of the patients.**

| Diagnostic test | No./No. (%) | Gender, No./No. (%) | | P value | Age, No./No. (%) | | P value |
|---|---|---|---|---|---|---|---|
| | | Male | Female | | 18–40 yr | 41–50 yr | |
| **Standard/routine blood tests** | | | | | | | |
| Hemoglobin >16.5 g/dL | 19/242 (7.9) | 18 (12.7) | 1 (1.0) | <0.001 | 6 (6.5) | 13 (8.7) | 0.55 |
| Platelet count >450,000 cells/μL | 12/242 (5.0) | 6 (4.2) | 6 (6.0) | 0.53 | 6 (6.5) | 6 (4.0) | 0.38 |
| Fasting blood glucose ≥126 mg/dL | 55/235 (23.4) | 34 (24.5) | 21 (21.9) | 0.65 | 18 (20.5) | 37 (25.2) | 0.41 |
| Hemoglobin A1C ≥6.5% | 47/130 (36.2) | 30 (36.1) | 17 (36.2) | 1.00 | 12 (30.0) | 35 (38.9) | 0.33 |
| LDL-C >130 mg/dL | 96/210 (45.7) | 64 (50.0) | 32 (39.0) | 0.12 | 34 (46.0) | 62 (45.6) | 0.96 |
| **Advanced blood tests** | | | | | | | |
| ESR >20 mm/hr | 18/53 (34.0) | 5 (19.2) | 13 (48.2) | 0.04 | 8 (26.7) | 10 (43.5) | 0.20 |
| Anti-HIV | 7/117 (6.0) | 2/70 (2.9) | 5/47 (10.6) | 0.12 | 3/69 (4.4) | 4/48 (8.3) | 0.44 |
| ANA titer | 14/58 (24.1) | 2/24 (8.3) | 12/34 (35.3) | 0.03 | 11/38 (29.0) | 3/20 (15.0) | 0.34 |
| Protein C deficiency | 3/51 (5.9) | 1/25 (4.0) | 2/26 (7.7) | 1.00 | 3/32 (9.4) | 0/19 (0) | 0.29 |
| Protein S deficiency | 6/51 (11.8) | 0/25 (0) | 6/26 (23.1) | 0.02 | 5/32 (15.6) | 1/19 (5.3) | 0.39 |
| Antithrombin III deficiency | 1/44 (2.3) | 1/21 (4.8) | 0/23 (0) | 0.48 | 1/27 (3.7) | 0/17 (0) | 1.00 |
| Lupus anticoagulant | 9/83 (10.8) | 3/33 (9.1) | 6/50 (12.0) | 1.00 | 4/54 (7.4) | 5/29 (17.2) | 0.27 |
| Anticardiolipin IgG | 1/83 (1.2) | 0/34 (0) | 1/49 (2.0) | 1.00 | 0/55 (0) | 1/28 (3.6) | 0.34 |
| Anti-beta-2-glycoprotein | 5/27 (18.5) | 0/5 (0) | 5/22 (22.7) | 0.55 | 2/18 (11.1) | 3/9 (33.3) | 0.30 |
| **Cardiac evaluation** | | | | | | | |
| TTE | 35/109 (32.1) | 19/64 (29.7) | 16/45 (35.6) | 0.52 | 18/53 (34.0) | 17/56 (30.4) | 0.69 |
| TEE | 6/10 (60.0) | 4/6 (66.7) | 2/4 (50.0) | 1.00 | 3/5 (60.0) | 3/5 (60.0) | 1.00 |
| 24-h Holter | 1/23 (4.4) | 0/14 (0) | 1/9 (11.1) | 0.39 | 1/10 (10.0) | 0/13 (0) | 0.44 |
| **Vascular study** | | | | | | | |
| CDUS | 1/5 (20.0) | 1/4 (25.0) | 0/1 (0) | 1.00 | 0/0 (NA) | 1/5 (20.0) | NA |
| CTA | 65/81 (80.3) | 38/49 (77.6) | 27/32 (84.4) | 0.57 | 30/41 (73.2) | 35/40 (87.5) | 0.16 |
| MRA | 30/69 (43.5) | 15/31 (48.4) | 15/38 (39.5) | 0.46 | 13/28 (46.4) | 17/41 (41.5) | 0.68 |
| **Specialized evaluation** | | | | | | | |
| Genetic testing (CADASIL, MELAS) | 3/3 (100.0) | 2/2 (100.0) | 1/1 (100.0) | 1.00 | 3/3 (100.0) | 0/0 (NA) | NA |

**Abbreviations:** ANA, antinuclear antibody; CADASIL, Cerebral Autosomal Dominant Arteriopathy with Sub-cortical Infarcts and Leukoencephalopathy; CDUS, carotid doppler ultrasonography; CTA, computed tomography angiography; ESR, erythrocyte sedimentation rate; HIV, human immunodeficiency virus; IgG, immunoglobulin G; LDL-C, low-density lipoprotein cholesterol; MELAS, Mitochondrial Encephalopathy, Lactic Acidosis, and Stroke-like episodes; MRA, magnetic resonance angiography; NA, not applicable; TEE, transesophageal echocardiography; TTE, transthoracic echocardiography.

out to determine the cause of the stroke, enhancing the yield of CTA diagnoses [9]. Although TEE has 60% positive test results for patients who were evaluated cardiac sources of stroke or emboli, it was only carried out in 4% of patients in our cohort and might not be an inference to its higher diagnostic yields when compared with other cardiac assessments. This could be explained by the fact that TTE is typically followed by TEE to offer a comprehensive evaluation of abnormal cardiogenic morphology [15, 16]. These noninvasive diagnostic procedures, however, are probably recommended for young individuals due to increasing the detection of vascular dissection and cardiac sources of stroke, especially PFO with or without ASA, which are the common causes of stroke among young adult patients.

Our result shows that the most common stroke subtype in the young, according to the TOAST classification of subtypes of acute ischemic stroke, was a stroke of other determined etiology, 31% in total. Similar epidemiologic data were also reported in Smajlović D.'s study, which found that strokes of other determined etiology and undetermined etiology collectively account for most stroke cases in young adults [2]. In comparison among the two age groups,

the highest etiology in age 18–40 is a stroke of other determined etiology, and the highest in age 41–50 is small-vessel occlusion. In contrast to the elderly, atrial fibrillation (AF), a common cause of ischemic stroke in older people, is less frequently observed in our cohort (5.7%). The proportions of cardioembolism and stroke with other determined etiologies declined as young adults aged, whereas the proportions of large-artery atherosclerosis and small-vessel occlusion increased [6]. The correlation of vascular risk factors, commonly found in aged individuals, and atherosclerotic-related stroke subtypes might be the reasons for the difference in stroke etiology between the two age groups [17]. Compared to patients aged 18 to 40, the prevalence of diabetes mellitus and dyslipidemia was significantly higher in the 41 to 50-year-old patient group. The prevalence of diabetes mellitus, hypertension, dyslipidemia, and obesity was higher in older age groups. Similar to a study by Sarnowski BV *et al.*, patients 45 years of age or older had higher rates of the majority of risk variables [18]. Smoking and alcohol drinking was shown to be highly associated with a higher proportion of the male sex, as shown in our study. One potential reason is that men may be more likely to engage in risky behaviors due to societal expectations and cultural norms.

In the subtype of the other determined etiology, the most common cause was the hypercoagulable state and vascular dissection, which account for 8.2 and 7.8 percent of other determined etiology, which was not significantly high due to the heterogeneity of other determined causes in young adults. Arterial hypercoagulable state testing, including APS antibodies testing, is preferable to perform than venous occlusion, with the positive result in up to 14% of cases overall [19]. Cervical artery dissection can cause about 7.8–20% of strokes in young adults, while internal carotid artery dissection is slightly higher than posterior circulation dissection. Ethnicity might affect the site of vascular dissection where vertebral artery dissection commonly occurs in Asian patients [20].

The 90-day functional outcomes measured using the modified Rankin Scale scoring systems of the younger group tended to be superior, represented by more patients with favorable outcomes (mRS 0–2) despite the lack of statistical significance due to the scarce number of patients. Aslam A. *et al.*'s study also has a similar tendency of excellent functional outcomes in younger patients compared to elderly strokes, emphasizing the essential role of determination of stroke etiology for the most appropriate approach to secondary stroke prevention [21].

In young individuals, strokes may be more challenging to diagnose due to atypical symptoms or a lack of awareness among healthcare providers about the possibility of stroke in this population. Our study could help identify factors contributing to missed or delayed diagnoses and inform strategies to improve the accuracy and timeliness of stroke diagnosis in this age group. It is imperative to underline the significance of etiological explication in determining the cause of a stroke, particularly among young individuals. It is recommended that noninvasive diagnostic tests, such as Holter monitoring and TTE, be conducted generally in young stroke patients as part of the diagnostic workup for stroke etiology. Additionally, such a study could help to identify potential risk factors for stroke in young individuals and inform the development of preventative measures to reduce the incidence of stroke in this population. Overall, a research study on the diagnosis of stroke in the young could have significant clinical benefits, including improved patient outcomes and a reduction in the burden of stroke on society.

We acknowledge some limitations to our study. First, only 36% of our patients received a full complement of diagnostic testing, and the absence of disease for those patients with incomplete workups could not be guaranteed. In order to incorporate data to conduct a large cohort, we analyzed data from the entire cohort even though there are some missing data due to a retrospective study. Second, there was selection bias for some specialized evaluation, namely genetic testing, because this diagnostic workup was customarily conducted in patients

with clinical clues or following other extensive studies. Lastly, because some diagnostic work-ups were performed in a small number in our cohort, such as TEE, representing a thin data-base of those investigations, we cannot state a conclusion on the yield of diagnostic tests or the value of TEE, especially in comparison with TTE in young adults.

## Conclusions

In conclusion, our study has shown Anti-HIV, TTE, hypercoagulable state, particularly APS testing, and noninvasive vascular investigation with CTA are typically conducted in addition to traditional assessments or basic laboratory testing for identifying stroke in young adults. Using these diagnostic tests incorporated with initial investigations following standard or rou-tine blood tests to determine stroke etiologies in properly selected young adult stroke or TIA patients can lead to a clinical benefit, as early diagnosis and treatment can significantly improve patient outcomes. Hypercoagulable state, noninflammatory vasculopathy, and poten-tially cardiac sources of stroke are common stroke etiologies among young individuals. In addition, primary and secondary stroke prevention is encouraged in these patients to establish more favorable functional outcomes. Further research is needed to determine the optimal use of these tests in different clinical settings and to identify any potential biases or limitations.

## Supporting information

**S1 Fig. Diagnostic tests of ischemic stroke in young adults.** ANA, antinuclear antibody; CDUS, carotid doppler ultrasonography; CTA, computed tomography angiography; ESR, erythrocyte sedimentation rate; HIV, human immunodeficiency virus; MRA, magnetic reso-nance angiography; TEE, transesophageal echocardiography; TTE, transthoracic echocardiog-raphy.
(TIF)

## Acknowledgments

We would like to thank the medical statisticians of the Research Unit Medicine (RUM) Department of Internal Medicine for guidance and assistance in statistical analysis.

## Author Contributions

**Conceptualization:** Jakraphat Sakseranee, Peerapong Sethabouppha, Tanapat Pattarasakulchai, Theerawee Klaewkla, Kitti Thiankhaw.

**Data curation:** Jakraphat Sakseranee, Peerapong Sethabouppha, Tanapat Pattarasakulchai, Theerawee Klaewkla, Kitti Thiankhaw.

**Formal analysis:** Kitti Thiankhaw.

**Investigation:** Jakraphat Sakseranee, Peerapong Sethabouppha, Tanapat Pattarasakulchai, Theerawee Klaewkla, Kitti Thiankhaw.

**Methodology:** Jakraphat Sakseranee, Peerapong Sethabouppha, Tanapat Pattarasakulchai, Theerawee Klaewkla, Kitti Thiankhaw.

**Project administration:** Kitti Thiankhaw.

**Supervision:** Kitti Thiankhaw.

**Visualization:** Kitti Thiankhaw.

**Writing – original draft:** Jakraphat Sakseranee, Peerapong Sethabouppha, Tanapat Pattarasakulchai, Theerawee Klaewkla.

**Writing – review & editing:** Kitti Thiankhaw.

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
