## [Decision Letter · Decision Letter 0]

10 May 2023

PONE-D-23-05766The Yield of Diagnostic Tests and Functional Outcomes of Ischemic Stroke or Transient Ischemic Attack in Young AdultsPLOS ONE

Dear Dr. Thiankhaw,

Thank you for submitting your manuscript to PLOS ONE. After careful consideration, we feel that it has merit but does not fully meet PLOS ONE’s publication criteria as it currently stands. Therefore, we invite you to submit a revised version of the manuscript that addresses the points raised during the review process.

We look forward to receiving your revised manuscript.

Kind regards,

Wolfgang Rudolf Bauer, M.D., Ph.D.

Academic Editor

PLOS ONE

Reviewers' comments:

Reviewer's Responses to Questions

**Comments to the Author**

1. Is the manuscript technically sound, and do the data support the conclusions?

Reviewer #1: Partly

Reviewer #2: Partly

2. Has the statistical analysis been performed appropriately and rigorously? 

Reviewer #1: I Don't Know

Reviewer #2: N/A

3. Have the authors made all data underlying the findings in their manuscript fully available?

Reviewer #1: Yes

Reviewer #2: Yes

4. Is the manuscript presented in an intelligible fashion and written in standard English?

Reviewer #1: Yes

Reviewer #2: Yes

5. Review Comments to the Author

Reviewer #1: Sakseranee et al report on an investigation of young and middle aged patients with stroke or TIA. One focus laid on assessment of causes of stroke and which diagnostic examination should be performed due to secondary prevention. They compared a younger (18-40y) and middle aged (41-50y) group. They described different causes of stroke between these groups. They stated genetic testing, CTA and TEE are the examinations with the most diagnostic value.

This is an interesting topic but many concerns remain:

1. Genetic testing was performed in only 3 pts. A general conclusion about its value could not be concluded and should not be so emphasized.

2. Complete diagnostic data were only available in 36%. This is a thin database to state conclusions.

3. In Conclusion: Stroke could not be diagnosed with TEE. Only causes of stroke could be evaluated.

4. Table 1:

a. Is a cause why congestive heart failure in the younger group is more often than in the older?

5. Table 2:

a. In the line with Ischemic stroke classification: to which belongs the p-value 0.001?

b. No p-values are given for: large artery arterosclerosis, cardioembolism, small vessel occlusion, stroke from other determined etiology and stroke of indetermined etiology.

c. There are patients, who didn´t get a TTE or TEE. Which type of diagnostic workup did they get?

Only in 10 patients a TEE was performed. Why? This is a small database to conclude anything about the value of TEE and especially in comparison with TTE.

Reviewer #2: The authors describe a large cohort of consecutive young adults with ischemic stroke or transient ischemic attack.The differentiation between younger (18-40 yr) and middle-aged patients (40-50 yrs) according to different clinical parameters, including the TOAST classification of subtypes, is scientifically interesting. Also, the presentation of the functional outcome of the patients is sound. However, one main aspect of the publication („the yield of diagnostic tests“) is insufficient, because these test have not been systematically performed (the Authors adequately mention the main limitation of their study: „A standardized stroke workup was not done“).

Therefore, the paper need major revision for a probable publication: The aspect „yield of diagnostic tests“ should be completely omitted. CT, genetic tests, TTE, TEE have been so rarely performed that no valid conclusion about the diagnostic yield of these tests can be drawn. The final conclusion in the abstract “Our study show a high-yield diagnostic test of CTA and TEE for deteremining stroke etiology in young adults. Prompt diagnosis and treatment of stroke can lead to better patient outcomes, making using these tests a valuable strategy“ is completely unproven and should be omitted.

In the „Introduction“ several statements are again pure speculative and should be omitted.

Figure 4 should be omitted!

6. PLOS authors have the option to publish the peer review history of their article (what does this mean?). If published, this will include your full peer review and any attached files.

Reviewer #1: No

Reviewer #2: No

---

## [Author Response · Author response to Decision Letter 0]

17 May 2023

Manuscript number: PONE-D-23-05766

Title: The diagnostic tests and functional outcomes of acute ischemic stroke or transient ischemic attack in young adults: A 4-year hospital-based observational study

Responses to Reviewers 

We would like to thank the editor and the reviewers for all the valuable comments and suggestions. We have revised the manuscript based on these constructive suggestions. We believe the revised manuscript is now greatly improved after the revisions have been made, as suggested by the reviewers and the editor.

Responses to Editorial Corrections

Q1: Please ensure that your manuscript meets PLOS ONE's style requirements, including those for file naming.

A1: We revised the manuscript and file naming following PLOS ONE’s style requirement.

Q2: We note that you have indicated that data from this study are available upon request. PLOS only allows data to be available upon request if there are legal or ethical restrictions on sharing data publicly.

A2: We would like to thank the editor for the correction. We revised Data Availability Statement based on the information on unacceptable data access restrictions. Now it read “Data cannot be shared publicly because of ethical issues. Data are available from the Ethics Committee of the Faculty of Medicine, Chiang Mai University (contact via Researchmed@cmu.ac.th) for researchers who meet the criteria for access to confidential data.” (In revised cover letter)

Responses to Reviewer 1:

Q1: Genetic testing was performed in only 3 pts. A general conclusion about its value could not be concluded and should not be so emphasized.

A1: We would like to thank the reviewer for the suggestion. We have revised the general conclusion regarding the value of genetic testing in young stroke patients, as suggested by the reviewer. Now it read “Conclusions: Stroke of other determined etiology remained the common cause of stroke in young adults, and most affected individuals had excellent clinical outcomes. Blood tests for arterial hypercoagulability and noninvasive vascular and cardiac evaluations are encouraged in selected patients to determine the stroke etiology and guide for appropriate preventive strategies.” (Pages 2-3, lines 38-41)

Q2: Complete diagnostic data were only available in 36%. This is a thin database to state conclusions.

A2: We would like to thank the reviewer for the valuable suggestion. We rearranged the conclusion of the revised manuscript to be more concise and precise according to a reviewer’s suggestion. Now it read “In conclusion, our study has shown Anti-HIV, TTE, hypercoagulable state, particularly APS testing, and noninvasive vascular investigation with CTA are typically conducted in addition to traditional assessments or basic laboratory testing for identifying stroke in young adults. Using these diagnostic tests incorporated with initial investigations following standard or routine blood tests to determine stroke etiologies in properly selected young adult stroke or TIA patients can lead to a clinical benefit, as early diagnosis and treatment can significantly improve patient outcomes.” (Pages 23-24, lines 311-316)

Q3: In Conclusion: Stroke could not be diagnosed with TEE. Only causes of stroke could be evaluated.

A3: We would like to thank the reviewer for the valuable suggestion. We rearranged the conclusion of the revised manuscript to be more concise and precise according to a reviewer’s suggestion. Now it read “Conclusions: Stroke of other determined etiology remained the common cause of stroke in young adults, and most affected individuals had excellent clinical outcomes. Blood tests for arterial hypercoagulability and noninvasive vascular and cardiac evaluations are encouraged in selected patients to determine the stroke etiology and guide for appropriate preventive strategies.” (Pages 2-3, lines 38-41) and “In conclusion, our study has shown Anti-HIV, TTE, hypercoagulable state, particularly APS testing, and noninvasive vascular investigation with CTA are typically conducted in addition to traditional assessments or basic laboratory testing for identifying stroke in young adults. Using these diagnostic tests incorporated with initial investigations following standard or routine blood tests to determine stroke etiologies in properly selected young adult stroke or TIA patients can lead to a clinical benefit, as early diagnosis and treatment can significantly improve patient outcomes.” (Pages 23-24, lines 311-316)

Q4: Table 1: Is a cause why congestive heart failure in the younger group is more often than in the older?

A4: We would like to thank the reviewer for the suggestion. The causes of congestive heart failure in the younger group are currently mentioned in the Results part. Now it read “Interestingly, congestive heart failure (CHF) was commonly diagnosed in the younger patient group, in which non-ischemic dilated cardiomyopathy was a certain cause in this population (10.8% vs. 1.3%, P = 0.001). Table 1 demonstrates the baseline characteristics and risk factors of the participants, classified into two cohorts.” (Page 8, lines 133-136)

A5: Table 2: In the line with Ischemic stroke classification: to which belongs the p-value 0.001?

Q5: We would like to thank the reviewer for the helpful suggestion. The P-value 0.001 belongs to the difference among all ischaemic stroke classifications. We performed an additional statistical analysis to test the difference among ischaemic stroke subtypes. Now it read “Compared to a younger age group, AIS aged 41-50 had more prevalence of large-artery atherosclerosis (LAA) and small-vessel occlusion (SVO) stroke subtype according to TOAST classification, 21.9% vs. 9.7%, P = 0.014 and 27.8% vs. 11.8%, P = 0.003, respectively (Fig 2).” (Page 10, lines 145-148) and Table 2 Pages 11-12.

Q6: No p-values are given for: large artery arterosclerosis, cardioembolism, small vessel occlusion, stroke from other determined etiology and stroke of indetermined etiology.

A6: We apologise for this mistake. We performed statistical analysis to test the difference among ischaemic stroke subtypes, as the reviewer suggested. Now it read “Focus on younger patients, AIS patients in the 18-40 age group had a significantly higher proportion of stroke of other determined etiology, 39.8% vs. 25.2%, P = 0.016, compared to an older group (Table 2 and Fig 2). The most common cause of ischemic stroke of other determined etiology is a hypercoagulable state (8.2%), and hereditary thrombophilia is commonly diagnosed (4.1%). These findings are associated with a higher proportion of anticoagulant prescriptions in AIS patients in the younger age group (23.7% vs. 13.4%, P = 0.04).” (Pages 13-14, lines 168-173) and Table 2 Pages 11-12.

Q7: There are patients, who didn´t get a TTE or TEE. Which type of diagnostic workup did they get? Only in 10 patients a TEE was performed. Why? This is a small database to conclude anything about the value of TEE and especially in comparison with TTE.

A7: We would like to thank the reviewer for the valuable suggestion. The information regarding TEE and additional discussion on its diagnostic value have been illustrated in the Results and Discussion part of the revised manuscript. Now it read “Among cardiac evaluation, transthoracic echocardiography (TTE) was commonly performed in 44.7% (109 of 244) of patients and provided positive results in 32.1% with no difference between gender or age groups. Transesophageal echocardiography (TEE) illustrated diagnostic results in 60% of included patients. Still, its diagnostic yield cannot be generalized clinical implication because it was performed on only ten patients and selected populations with highly suspected positive cardiac causes.” (Page 16, lines 201-206) and “Although TEE has 60% positive test results for patients who were evaluated cardiac sources of stroke or emboli, it was only carried out in 4% of patients in our cohort and might not be an inference to its higher diagnostic yields when compared with other cardiac assessments. This could be explained by the fact that TTE is typically followed by TEE to offer a comprehensive evaluation of abnormal cardiogenic morphology.[15, 16] These noninvasive diagnostic procedures, however, are probably recommended for young individuals due to increasing the detection of vascular dissection and cardiac sources of stroke, especially PFO with or without ASA, which are the common causes of stroke among young adult patients.” (Page 21, lines 257-264)

Responses to Reviewer 2:

Q1: However, one main aspect of the publication (“the yield of diagnostic tests”) is insufficient, because these test have not been systematically performed (the Authors adequately mention the main limitation of their study: “A standardized stroke workup was not done”). The aspect “yield of diagnostic tests” should be completely omitted.

A1: We would like to thank the reviewer for the valuable suggestion. We decided to adjust the study title for suitability and appropriateness with the details of the revised manuscript. Now it read “The diagnostic tests and functional outcomes of acute ischemic stroke or transient ischemic attack in young adults: A 4-year hospital-based observational study” (Page 1, lines 1-2) In addition, the Abstract and Discussion parts have been rearranged by adding additional scientific information to enhance the quality of the work. Now it read “Background and objectives: Ischemic strokes in young adults have been a significant concern due to various potential etiologies and had substantial clinical and public health impacts. We aimed to study the diagnostic tests, etiologies, and functional outcomes of acute ischemic stroke (AIS) and transient ischemic attack (TIA) in young adult patients.

Methods: The data were retrieved from the Chiang Mai University Hospital Stroke Registry between January 2018 and December 2021. Consecutive AIS or TIA patients were included if they were 18-50 years and had no stroke mimics. Study outcomes were proportions of positive diagnostic tests, and 90-day modified Rankin Scale (mRS).

Results: Of 244 enrolled patients, 59.0% (n = 144) were male, and 38.1% (n = 93) were aged 18-40, classified as the younger age group. There was a high incidence of diabetes (24.5%) and dyslipidemia (54.3%) among patients aged 41-50, associated with small-vessel occlusion and large-artery atherosclerosis stroke classification in this age group. Patients aged 18-40 years had more other determined etiologies (39.8%), with hypercoagulability (8.2%), arterial dissection (7.8%), and cardiac sources (6.6%) being the first three causes, which were associated with higher anticoagulant treatment. The cerebrovascular study, cardiac evaluation using echocardiography, and antiphospholipid syndrome testing were commonly performed, of which computed tomography angiography provided a high proportion of positive results (80.3%). 76.3% of young adult patients had excellent functional outcomes (mRS 0-1) with a median mRS of 0 (interquartile range 0-1) at 90-day follow-up.

Conclusions: Stroke of other determined etiology remained the common cause of stroke in young adults, and most affected individuals had excellent clinical outcomes. Blood tests for arterial hypercoagulability and noninvasive vascular and cardiac evaluations are encouraged in selected patients to determine the stroke etiology and guide for appropriate preventive strategies.” (Pages 2-3, lines 19-41), “On the contrary, the diagnostic yields of these investigations, especially compared to others, should be interpreted cautiously because it was not entirely performed in all cohorts.” (Page 20, lines 230-232), and “The specialized investigation, genetic testing for CADASIL (NOTCH3 mutation) and MELAS is often conducted in young adult patients with clinical clues compatible with these stroke syndromes. Additionally, these diagnostic procedures are usually done as a step-by-step approach in patients who have undergone all other investigations yet have not found any etiology, which only included three patients in our study. As a result, the value of these diagnostic yields typically varied between the studies. Dong Y.’s study conducted in two hundred eighteen consecutive patients found the overall frequency of CADASIL mutations in lacunar stroke to be less than 0.5%.[11] Therefore, the difference in the suspected population might affect the inconsistent results of the diagnostic yield of genetic tests in young individuals who suffer from acute ischemic stroke or TIA.

The vascular study of cerebral vessels and neck, including CTA, magnetic resonance angiography (MRA), or carotid doppler ultrasonography (CDUS), provides details of cerebral vasculatures and guides therapeutic interventions in acute stroke settings and long-term secondary stroke prevention.[12, 13] Therefore, noncontrast computed tomography (NCCT) and CTA brain have been utilized as standard stroke neuroimaging in acute stroke patients.[14] Among these investigations, our finding reveals that CTA represents a higher proportion of positive results compared to MRA and CDUS. Limitations and accessibility of MRA and operator-dependent of CDUS might be the answer for our cohort that these vascular imagings are lower number performed and proportion of positive tests. Similar to Ji R. et al., which also found a superiority of vascular imaging, CTA, in young individuals.[15]” (Pages 20-21, lines 234-253)

Q2: The final conclusion in the abstract “Our study show a high-yield diagnostic test of CTA and TEE for deteremining stroke etiology in young adults. Prompt diagnosis and treatment of stroke can lead to better patient outcomes, making using these tests a valuable strategy” is completely unproven and should be omitted.

A2: We would like to thank the reviewer for the helpful suggestion. As the reviewer suggested, the conclusion of the revised manuscript has been rewritten to be more concise and precise. Now it read “Conclusions: Stroke of other determined etiology remained the common cause of stroke in young adults, and most affected individuals had excellent clinical outcomes. Blood tests for arterial hypercoagulability and noninvasive vascular and cardiac evaluations are encouraged in selected patients to determine the stroke etiology and guide for appropriate preventive strategies.” (Pages 2-3, lines 38-41) and “In conclusion, our study has shown Anti-HIV, TTE, hypercoagulable state, particularly APS testing, and noninvasive vascular investigation with CTA are typically conducted in addition to traditional assessments or basic laboratory testing for identifying stroke in young adults. Using these diagnostic tests incorporated with initial investigations following standard or routine blood tests to determine stroke etiologies in properly selected young adult stroke or TIA patients can lead to a clinical benefit, as early diagnosis and treatment can significantly improve patient outcomes.” (Pages 23-24, lines 311-316)

Q3: In the “Introduction” several statements are again pure speculative and should be omitted.

A3: We would like to thank the reviewer for the suggestion. The Introduction part has been revised to clearly state the unknown and why it is important to know. Now it read “Strokes in young adult patients are considered not uncommon, accounting for about 15% of all stroke patients.[1] It significantly affects the quality of life and has high economic impacts since affected individuals might become disabled before their most productive years, in contrast to strokes in older persons.[2] This background makes preventing stroke in young adults an urgent global public health concern. Because the etiologic spectrum of ischemic stroke in young adults is broader and more heterogeneous than in older people,[3] several investigations and tests are usually performed to evaluate all potential etiologies to select appropriate secondary stroke prevention, including standard, advanced, and specialized evaluation.[4]

Previous studies have been conducted to elucidate the risk factors, stroke outcomes and its predictors, and causes of stroke in young adult patients with the general agreement that uncommon causes and embolic sources of stroke might be the potential causes in individuals with younger age.[5-7] Importantly, there is no consensus on the definition of stroke in the young, and these studies ranged in age from 40 to 60 while determining young adults. Additionally, studies focused on different causes between gender and compared among young and younger adults are limited.

In the present study, we aimed to study the diagnostic tests of acute ischemic stroke (AIS) and transient ischemic attack (TIA) in young adults. We also investigated the causes and functional outcomes of stroke in this population using a large cohort of consecutive young adults with AIS or TIA. We hypothesized that our findings probably guided effective diagnostic methods to reduce the number of unnecessary investigations and establish appropriate stroke prevention.” (Page 4, lines 46-64)

Q4: Figure 4 should be omitted!

A4: We would like to thank the reviewer for the suggestion. Figure 4 is currently moved to a supporting information. Now it read “S1 Fig. Diagnostic tests of ischemic stroke in young adults. ANA, antinuclear antibody; CDUS, carotid doppler ultrasonography; CTA, computed tomography angiography; ESR, erythrocyte sedimentation rate; HIV, human immunodeficiency virus; MRA, magnetic resonance angiography; TEE, transesophageal echocardiography; TTE, transthoracic echocardiography.” (Page 30, lines 401-404) and S1_Fig.

---

## [Decision Letter · Decision Letter 1]

1 Sep 2023

PONE-D-23-05766R1The diagnostic tests and functional outcomes of acute ischemic stroke or transient ischemic attack in young adults: A 4-year hospital-based observational studyPLOS ONE

Dear Dr. Thiankhaw,

Thank you for submitting your manuscript to PLOS ONE. After careful consideration, we feel that it has merit but does not fully meet PLOS ONE’s publication criteria as it currently stands. Therefore, we invite you to submit a revised version of the manuscript that addresses the points raised during the review process.

ACADEMIC EDITOR: The authors are thanked for this submission to PLOS ONE. After a critical external peer review by three experts, I reinforced improving your paper's clarity and presentation and acknowledging study limitations according to the reviewer's concerns. Please see the attached reviewer comments detail below.

We look forward to receiving your revised manuscript.

Kind regards,

Redoy Ranjan, MBBS, MRCSEd, Ch.M., MS (CV&TS), FACS

Academic Editor

PLOS ONE

Journal Requirements:

Reviewers' comments:

Reviewer's Responses to Questions

**Comments to the Author**

1. If the authors have adequately addressed your comments raised in a previous round of review and you feel that this manuscript is now acceptable for publication, you may indicate that here to bypass the “Comments to the Author” section, enter your conflict of interest statement in the “Confidential to Editor” section, and submit your "Accept" recommendation.

Reviewer #1: All comments have been addressed

Reviewer #3: All comments have been addressed

Reviewer #4: All comments have been addressed

2. Is the manuscript technically sound, and do the data support the conclusions?

Reviewer #1: Partly

Reviewer #3: Yes

Reviewer #4: Yes

3. Has the statistical analysis been performed appropriately and rigorously? 

Reviewer #1: I Don't Know

Reviewer #3: Yes

Reviewer #4: Yes

4. Have the authors made all data underlying the findings in their manuscript fully available?

Reviewer #1: Yes

Reviewer #3: Yes

Reviewer #4: Yes

5. Is the manuscript presented in an intelligible fashion and written in standard English?

Reviewer #1: Yes

Reviewer #3: Yes

Reviewer #4: Yes

6. Review Comments to the Author

Reviewer #1: The revised manuscript by Sakseranee et al improved and the authors tried to answer the questions and changed the manuscript.

Although some concerns remain:

The quality of data is limited. Therefore, a section “Limitations of the study” should be added. Here, all limitiations of the study should be mentioned.

Again, only the patients with complete data set should be analyzed, or minimum all patients with TTE.

How often was atrial fibrillation – a common cause for stroke - found in the cohort?

Table 1 and Table 5 do not clearly show sex-distribution within the subgroups.

In general, it should be more emphasized, that the etiological clarification of the cause of a stroke especially in young patients is very important. Non-invasive tests like TTE and holter should be performed in ALL patients to get an explanation for a stroke.

Reviewer #3: The authors examined the diagnostic tests, etiologies, and functional outcomes of acute ischemic stroke (AIS) and transient ischemic attack (TIA) in young patients in a single-center study of 244 patients over a 3-year period. In patients aged 18-40 years, the main reason was found to be "other determined etiologies" with 39.8%. The article is well prepared methodologically and linguistically. It is recommended to accept

Reviewer #4: I do appreciate the rewriting process conducted according to previous reviews as stroke in young is not uncommon problem and thus further publications are warranted. TEE, loop recorders and CTA utilization rate should be more encouraged in future publications.

7. PLOS authors have the option to publish the peer review history of their article (what does this mean?). If published, this will include your full peer review and any attached files.

Reviewer #1: No

Reviewer #3: **Yes: **Bilgehan Atılgan Acar

Reviewer #4: **Yes: **Mohamed Mostafa

---

## [Author Response · Author response to Decision Letter 1]

3 Sep 2023

Manuscript number: PONE-D-23-05766R1

Title: The diagnostic tests and functional outcomes of acute ischemic stroke or transient ischemic attack in young adults: A 4-year hospital-based observational study

Responses to Reviewers 

We would like to thank the academic editor and the reviewers for all the valuable comments and suggestions. We have revised the manuscript based on these constructive suggestions. We believe the revised manuscript is now greatly improved after the revisions have been made, as suggested by the reviewers and the editor.

Responses to Academic Editor

Q1: The authors are thanked for this submission to PLOS ONE. After a critical external peer review by three experts, I reinforced improving your paper's clarity and presentation and acknowledging study limitations according to the reviewer's concerns. Please see the attached reviewer comments detail below.

A1: We would like to thank the academic editor for the suggestion. The manuscript has been revised according to the reviewers’ comments.

Q2: Please review your reference list to ensure that it is complete and correct. If you have cited papers that have been retracted, please include the rationale for doing so in the manuscript text, or remove these references and replace them with relevant current references. Any changes to the reference list should be mentioned in the rebuttal letter that accompanies your revised manuscript. If you need to cite a retracted article, indicate the article’s retracted status in the References list and also include a citation and full reference for the retraction notice.

A2: We would like to thank the academic editor for the suggestion. The reference list has been thoroughly checked for its completeness and accuracy. No article cited in the manuscript has been retracted.

Responses to Reviewer #1:

Q1: The revised manuscript by Sakseranee et al improved and the authors tried to answer the questions and changed the manuscript. Although some concerns remain. The quality of data is limited. Therefore, a section “Limitations of the study” should be added. Here, all limitiations of the study should be mentioned.

A1: We would like to thank the reviewer for the helpful suggestions. We have revised the manuscript by adding a section on the limitations of the study, as suggested by the reviewer. Now it read “We acknowledge some limitations to our study. First, only 36% of our patients received a full complement of diagnostic testing, and the absence of disease for those patients with incomplete workups could not be guaranteed. In order to incorporate data to conduct a large cohort, we analyzed data from the entire cohort even though there are some missing data due to a retrospective study. Second, there was selection bias for some specialized evaluation, namely genetic testing, because this diagnostic workup was customarily conducted in patients with clinical clues or following other extensive studies. Lastly, because some diagnostic workups were performed in a small number in our cohort, such as TEE, representing a thin database of those investigations, we cannot state a conclusion on the yield of diagnostic tests or the value of TEE, especially in comparison with TTE in young adults.” (Pages 23-24, lines 311-320).

Q2: Again, only the patients with complete data set should be analyzed, or minimum all patients with TTE.

A2: We would like to thank the reviewer for the comment. In order to incorporate data to conduct a large cohort, we analyzed data from the entire cohort even though there are some missing data due to a retrospective study. We added this statement to the limitations of the study, as suggested by the reviewer.

Q3: How often was atrial fibrillation – a common cause for stroke - found in the cohort?

A3: We would like to thank the reviewer for the comment. We added the important issue regarding atrial fibrillation to the Discussion of the revised manuscript. Now it read “In contrast to the elderly, atrial fibrillation (AF), a common cause of ischemic stroke in older people, is less frequently observed in our cohort (5.7%).” (Pages 22, lines 271-271).

Q4: Table 1 and Table 5 do not clearly show sex-distribution within the subgroups.

A4: We would like to thank the reviewer for the comment. We acknowledge that gender disparity might affect the difference in stroke etiology between males and females. We, therefore, divided our cohort into two sub-cohorts, including gender and age cohorts (Fig 1). We revised the header row of Tables 1 and 5 for a more specific illustration, as suggested by the reviewer. (Table 1, Page 8-10 and Table 5, Page 17-19)

Q5: In general, it should be more emphasized, that the etiological clarification of the cause of a stroke especially in young patients is very important. Noninvasive tests like TTE and holter should be performed in ALL patients to get an explanation for a stroke.

A5: We would like to thank the reviewer for the valuable suggestion. We underscore the significance of the etiological elucidation of the cause of stroke, as suggested by the reviewer. Now it read “It is imperative to underline the significance of etiological explication in determining the cause of a stroke, particularly among young individuals. It is recommended that noninvasive diagnostic tests, such as Holter monitoring and TTE, be conducted generally in young stroke patients as part of the diagnostic workup for stroke etiology.” (Pages 23, lines 302-306).

Responses to Reviewer #3:

Q1: The authors examined the diagnostic tests, etiologies, and functional outcomes of acute ischemic stroke (AIS) and transient ischemic attack (TIA) in young patients in a single-center study of 244 patients over a 3-year period. In patients aged 18-40 years, the main reason was found to be "other determined etiologies" with 39.8%. The article is well prepared methodologically and linguistically. It is recommended to accept.

A1: We would like to thank the reviewer for the valuable comment. We sincerely appreciate your encouragement.

Responses to Reviewer #4:

Q1: I do appreciate the rewriting process conducted according to previous reviews as stroke in young is not uncommon problem and thus further publications are warranted. TEE, loop recorders and CTA utilization rate should be more encouraged in future publications.

A1: We would like to thank the reviewer for the valuable comment. We sincerely appreciate your encouragement.

---

## [Decision Letter · Decision Letter 2]

18 Sep 2023

The diagnostic tests and functional outcomes of acute ischemic stroke or transient ischemic attack in young adults: A 4-year hospital-based observational study

PONE-D-23-05766R2

Dear Dr. Thiankhaw,

We’re pleased to inform you that your manuscript has been judged scientifically suitable for publication and will be formally accepted for publication once it meets all outstanding technical requirements.

Kind regards,

Redoy Ranjan, MBBS, MRCSEd, Ch.M., MS (CV&TS), FACS

Academic Editor

PLOS ONE

Additional Editor Comments (optional):

Review Comments to the Author

Reviewer #1: The manuscript improved very much. All mentioned issues and limitations were included in the manuscript.

Reviewer #4: reference citations were optimized, which is considered an improved publication criterion. As discussed earlier, further publications are warranted. TEE, loop recorders and CTA utilization rate should be more encouraged in future publications.

---

## [Editor Report · Acceptance letter]

25 Sep 2023

PONE-D-23-05766R2 

The diagnostic tests and functional outcomes of acute ischemic stroke or transient ischemic attack in young adults: A 4-year hospital-based observational study 

Dear Dr. Thiankhaw:

I'm pleased to inform you that your manuscript has been deemed suitable for publication in PLOS ONE. Congratulations! Your manuscript is now with our production department. 

Kind regards, 

on behalf of

Dr. Redoy Ranjan 

Academic Editor

PLOS ONE